# EVIDENTIAL TURING PROCESSES

**Melih Kandemir**
Dept of Math and Computer Science
University of Southern Denmark
Odense, Denmark
`kandemir@imada.sdu.dk`

**Abdullah Akgül**
Department of Computer Engineering
Istanbul Technical University
Istanbul, Turkey
`akgula15@itu.edu.tr`

**Manuel Haussmann**[*]
Department of Computer Science
Aalto University
Espoo, Finland
`manuel.haussmann@aalto.fi`

**Gozde Unal**
Department of Computer Engineering
Istanbul Technical University
Istanbul, Turkey
`gozde.unal@itu.edu.tr`

## ABSTRACT

A probabilistic classifier with reliable predictive uncertainties i) fits successfully to the target domain data, ii) provides calibrated class probabilities in difficult regions of the target domain (e.g. class overlap), and iii) accurately identifies queries coming out of the target domain and rejects them. We introduce an original combination of Evidential Deep Learning, Neural Processes, and Neural Turing Machines capable of providing all three essential properties mentioned above for total uncertainty quantification. We observe our method on five classification tasks to be the only one that can excel all three aspects of total calibration with a single standalone predictor. Our unified solution delivers an implementation-friendly and compute efficient recipe for safety clearance and provides intellectual economy to an investigation of algorithmic roots of epistemic awareness in deep neural nets.

## 1 INTRODUCTION

The applicability of deep neural nets to safety-critical use cases such as autonomous driving or medical diagnostics is an active matter of fundamental research (Schwalbe & Schels, 2020). Key challenges in the development of total safety in deep learning are at least three-fold: i) evaluation of model fit, ii) risk assessment in difficult regions of the target domain (e.g. class overlap) based on which safety-preserving fallback functions can be deployed, and iii) rejection of inputs that do not belong to the target domain, such as an image of a cat presented to a character recognition system. The attention of the machine learning community to each of these critical safety elements is in steady increase (Naeini et al., 2015; Guo et al., 2017; Kuleshov et al., 2018). However, the developments often follow isolated directions and the proposed solutions are largely fragmented. Calibration of neural net probabilities focuses primarily on post-hoc adjustment algorithms trained on validation sets from in-domain data (Guo et al., 2017), ignoring the out-of-domain detection and model fit evaluation aspects. On the other hand, recent advances in out-of-domain detection build on strong penalization of divergence from the probability mass observed in the target domain (Sensoy et al., 2018; Malinin & Gales, 2018), distorting the quality of class probability scores. Such fragmentation of best practices not only hinders their accessibility by real-world applications but also complicates the scientific inquiry of the underlying reasons behind the sub-optimality of neural net uncertainties.

We aim to identify the guiding principles for Bayesian modeling that could deliver the most complete set of uncertainty quantification capabilities in one single model. We first characterize Bayesian models with respect to the relationship of their global and local variables: (a) *Parametric Bayesian Models* comprise a likelihood function that maps inputs to outputs via probabilistic global parameters, (b) *Evidential Bayesian Models* apply an uninformative prior distribution on the parameters of the likelihood function and infer the prior hyperparameters by amortizing on an input observation. Investigating the decomposition of the predictive distribution variance, we find out that the

---

[*]Work done while at Heidelberg University, Germany.

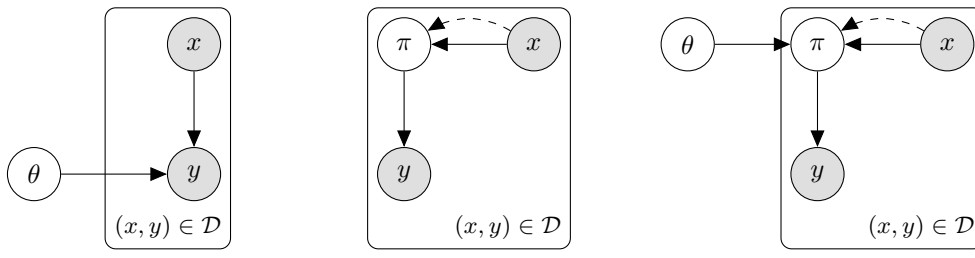

(a) Parametric Bayesian Model    (b) Evidential Bayesian Model    (c) Complete Bayesian Model

Figure 1: Plate diagrams of three main Bayesian modeling approaches. Shaded nodes are observed and unshaded ones are latent. The solid direct arrows denote conditional dependencies in the true model, while the bent dashed arrows denote amortization in variational inference.

uncertainty characteristics of these two approaches exhibit complementary strengths. We introduce a third approach that employs a prior on the likelihood parameters that both conditions on the input observations on the true model and amortizes on them during inference. The mapping from the input observations to the prior hyperparameters is governed by a random set of global parameters. We refer to the resulting approach as a (c) *Complete Bayesian Model*, since its predictive distribution variance combines the favorable properties of Parametric and Evidential Bayesian Models. Figure 1 gives an overview of the relationship between these three Bayesian modeling approaches.

The advantages of the Complete Bayesian Models come with the challenge of choosing an input-dependent prior. We introduce a new stochastic process construction that addresses this problem by accumulating observations from a context set during training time into the parameters of a global hyperprior variable by an explicitly designed update rule. We design the input-specific prior on the likelihood parameters by conditioning it on both the hyperprior and the input observation. As conditioning via explicit parameter updates amounts to maintaining an external memory that can be queried by the prior distribution, we refer to the eventual stochastic process as a *Turing Process* with inspiration from the earlier work on Neural Turing Machines (Graves et al., 2014) and Neural Processes (Garnelo et al., 2018b). We arrive at our target Bayesian design that is equipped with the complete set of uncertainty quantification capabilities by incorporating the Turing Process design into a Complete Bayesian Model. We refer to the resulting approach as an *Evidential Turing Process*. We observe on five real-world classification tasks that the Evidential Turing Process is the only model that excels simultaneously at model fit, class overlap quantification, and out-of-domain detection.

## 2 PROBLEM STATEMENT: TOTAL CALIBRATION

Consider the following data generating process with $K$ modes

$$y|\pi \sim \mathcal{C}at(y|\pi), \qquad x|y \sim \sum_{k=1}^{K} \mathbb{I}_{y=k} p(x|y=k), \tag{1}$$

where $\mathcal{C}at(y|\cdot)$ is a categorical distribution, $\mathbb{I}_u$ is the indicator function for predicate $u$, and $p(x|y)$ is the class-conditional density function on input observation $x$. For any prior class distribution $\pi$, the classification problem can be cast as identifying the class probabilities for observed patterns $Pr[y|x,\pi] = \mathcal{C}at(y|f_{true}^{\pi}(x))$ where $f_{true}^{\pi}(x)$ is a $K-$dimensional vector with the $k$-th element

$$f_{true}^{\pi,k}(x) = \pi_k p(x|y=k) \Big/ \sum_{\kappa=1}^{K} \pi_\kappa p(x|y=\kappa). \tag{2}$$

In many real-world cases $f_{true}^{\pi}$ is not known and should be identified from a hypothesis space $h_\pi \in \mathcal{H}_\pi$ via inference given a set of samples $\mathcal{D} = \{(x_n, y_n)|n = 1, \ldots, N\}$ obtained from the true distribution. The risk of the Bayes classifier $\arg\max_y \mathcal{C}at(y|h_\pi(x))$ for a given input $x$ is

$$R(h_\pi(x)) = \sum_{\kappa=1}^{K} \mathbb{I}_{\kappa \neq \arg\max h_\pi(x)} f_{true}^{\pi,\kappa}(x). \tag{3}$$

For the whole data distribution, we have $R(h_\pi) = \mathbb{E}_{x|\pi}[R(h_\pi(x))]$. The optimal case is when $h_\pi(x) = f^\pi_{true}(x)$, which brings about the irreducible Bayes risk resulting from class overlap:

$$R_* \left( f^\pi_{true}(x) \right) = \min \left\{ 1 - \max f^\pi_{true}(x), \max f^\pi_{true}(x) \right\}. \tag{4}$$

**Total Calibration.** Given a set of test samples $\mathcal{D}_{ts}$ coming from the true data generating process in Eq. 1, we define *Total Calibration* as the capability of a discriminative predictor $h_\pi(x)$ to quantify the following three types of uncertainty simultaneously:

*i) Model misfit* evaluated by the similarity of $h_\pi(x)$ and $f^\pi_{true}(x)$ measurable directly via

$$\text{KL}\big(\mathcal{C}at(y|f^\pi_{true}(x))p(x) \,||\, \mathcal{C}at(y|h_\pi(x))p(x)\big) = $$
$$\mathbb{E}_{p(x)} \left[ \log \mathcal{C}at(y|f^\pi_{true}(x) \right] + \mathbb{E}_{p(x)} \left[ -\log \mathcal{C}at(y|h_\pi(x)) \right]$$

where $\text{KL}(\cdot||\cdot)$ denotes the Kullback-Leibler divergence. Since $\mathbb{E}_{p(x)} \left[ \log \mathcal{C}at(y|f^\pi_{true}(x)) \right]$ is a constant with respect to $h_\pi(x)$, an unbiased estimator of the relative goodness of an inferred model is the negative test log-likelihood $NLL(h_\pi(x)) = -1/|\mathcal{D}_{ts}| \sum_{(x,y)\in\mathcal{D}_{ts}} \log \mathcal{C}at(y|h_\pi(x))$.

*ii) Class overlap* evaluated by $R_*(f^\pi_{true}(x))$. As there is no way to measure this quantity from $\mathcal{D}_{ts}$, it is approximated indirectly via *Expected Calibration Error (ECE)*

$$ECE[h_\pi] = \sum_{m=1}^{M} \frac{|B_m|}{N} \Big| acc(B_m) - conf(B_m) \Big|$$

where $B_m = \{(n|h_\pi(x_n) \in [(m-1)/M, m/M]\}$ are $M$ bins that partition the test set $\mathcal{D}_{ts}$ into $\mathcal{D}_{ts} = B_1 \cup \cdots \cup B_M$ and are characterized by their accuracy and confidence scores

$$acc(B_m) = \frac{1}{|B_m|} \sum_{n \in B_m} \mathbb{I}_{y_n = \arg\max h_\pi(x_n)}, \qquad conf(B_m) = \frac{1}{|B_m|} \sum_{n \in B_m} h_\pi(x_n). \tag{5}$$

*iii) Domain mismatch* defines the rarity of an input pattern $x_*$ for the target task, that is $Pr[p(x = x_*) < \epsilon]$ for small $\epsilon$. This quantity cannot be measured since $p(x)$ is unknown. It is approximated indirectly as the success of discriminating between samples $x_*$ coming from a different data distribution and those in $\mathcal{D}_{ts}$ by calculating the Area Under ROC (AUROC) curve w.r.t. $h_\pi(x)$.

## 3 UNCERTAINTY QUANTIFICATION WITH PARAMETRIC AND EVIDENTIAL BAYESIAN MODELS

**Parametric Bayesian Models.** A commonplace design choice is to build the hypothesis space by equipping the hypothesis function with a global parameter $\theta$ that takes values from a feasible set $\Theta$, that is $\mathcal{H}_\pi = \{h^\theta_\pi(x)|\theta \in \Theta\}$. *Parametric Bayesian Models (PBMs)* employ a prior belief $\theta \sim p(\theta)$, which is updated via Bayesian inference on a training set $\mathcal{D}_{tr}$ as $p(\theta|\mathcal{D}_{tr}) = \prod_{(x,y)\in\mathcal{D}_{tr}} \mathcal{C}at(y|h^\theta_\pi(x))p(\theta)/p(\mathcal{D}_{tr})$. Consider the update on the posterior belief

$$p(\theta|\mathcal{D}_{tr} \cup (x_*, y_*)) = \mathcal{C}at(y_*|h^\theta_\pi(x_*))p(\theta|\mathcal{D}_{tr})/Z, \tag{6}$$

after a new observation $(x_*, y_*)$ where the normalizer can be expressed as

$$Z = \frac{1}{p(\mathcal{D}_{tr})} \int \mathcal{C}at(y_*|h^\theta_\pi(x_*))L(\theta)p(\theta)d\theta. \tag{7}$$

with $L(\theta) = \prod_{(x,y)\in\mathcal{D}_{tr}} \mathcal{C}at(y|h^\theta_\pi(x))$. This can be seen as an inner product between the functions $\mathcal{C}at(y_*|h^\theta_\pi(x_*))$ and $L(\theta)$ on an embedding space modulated by $p(\theta)$. As the sample size grows, the high-density regions of $\mathcal{C}at(y_*|h^\theta_\pi(x_*))$ and $L(\theta)$ will superpose with higher probability, causing $Z$ to increase, making $p(\theta|\mathcal{D}_{tr} \cup (x_*, y_*))$ increasingly more peaked, and consequently we will have

$$\lim_{|\mathcal{D}_{tr}|\to+\infty} Var[\theta|\mathcal{D}_{tr}] = 0. \tag{8}$$

This conjecture depends on the assumption that a single random variable $\theta$ modulates all samples. The variance of a prediction in favor of a class $k$ decomposes according to the law of total variance

$$Var[y_* = k|x_*, \mathcal{D}] = \underbrace{Var_{\theta \sim p(\theta|\mathcal{D}_{tr})}[h_{\pi,k}^{\theta}(x_*)]}_{\text{Reducible model uncertainty}} + \underbrace{\mathbb{E}_{\theta \sim p(\theta|\mathcal{D}_{tr})}\left[(1 - h_{\pi,k}^{\theta}(x_*))h_{\pi,k}^{\theta}(x_*)\right]}_{\text{Data uncertainty}}. \quad (9)$$

Due to the Eq. 8, the first term on the r.h.s. vanishes in the large sample regime and the second one coincides with the asymptotic solution of the maximum likelihood estimator $\theta_{MLE}$. In cases when $\mathcal{H}_\pi$ contains $f_{true}^\pi(x)$, an ideal inference scheme has the potential to recover it and we get

$$\lim_{N \to +\infty} \mathbb{E}_{\theta \sim p(\theta|\mathcal{D}_{tr})}\left[(1 - h_{\pi,k}^{\theta}(x_*))h_{\pi,k}^{\theta}(x_*)\right] = (1 - h_{\pi,k}^{\theta_{MLE}}(x_*))h_{\pi,k}^{\theta_{MLE}}(x_*)$$
$$= (1 - f_{true}^\pi(x_*))f_{true}^\pi(x_*).$$

The simple proposition below sheds light on the meaning of this result. It points to the fact that the Bayes risk of a classifier (Eq. 4) and the second component of its predictive variance (Eq 9) are proportional. This gives a mechanistic explanation for why the second term on the r.h.s. of Eq. 9 quantifies class overlap. Despite the commonplace allocation of the wording *data uncertainty* for this term, we are not aware of prior work that derives it formally from basic learning-theoretic concepts.

**Proposition.** *The following inequality holds for any $\pi, \pi' \in [0.5, 1]$ pair*

$$\min(1 - \pi, \pi) \geq \min(1 - \pi', \pi') \Rightarrow (1 - \pi)\pi \geq (1 - \pi')\pi'.$$

*Proof.* Define $\pi = 0.5 + \epsilon$ and $\pi' = 0.5 + \epsilon'$ for $\epsilon, \epsilon' > 0$. As $\min(1 - \pi, \pi) \geq \min(1 - \pi', \pi') \Rightarrow \epsilon \leq \epsilon'$, we get $(1 - \pi)\pi = (1 - 0.5 - \epsilon)(0.5 + \epsilon) = 0.25 - 0.25\epsilon - \epsilon^2 \geq 0.25 - 0.25\epsilon' - \epsilon'^2 = (1 - \pi')\pi' \square$

The conclusion is that $(1 - f_{true}^\pi(x))f_{true}^\pi(x) \propto R_*(f_{true}^\pi(x))$. Hence, the second term on the r.h.s. of Eq. 9 is caused by the class overlap and it can be used to approximate the Bayes risk. Put together, the first term of the decomposition reflects the missing knowledge on the optimal value of $\theta$ that can be reduced by increasing the training set size, while the second reflects the uncertainty stemming from the properties of the true data distribution. Hence we refer to the first term as the *reducible model uncertainty* and the second the *data uncertainty*.

**Evidential Bayesian Models.** In all the analysis above, we assumed a fixed and unknown $\pi$ that modulates the whole process and cannot be identified by the learning scheme. When we characterize the uncertainty on the class distribution by a prior belief $\pi \sim p(\pi)$, we attain the joint $p(y, \pi|x) = Pr[y|x, \pi]p(\pi|x)$. The variance of the Bayes optimal classifier that averages over this uncertainty $Pr[y|x] = \int Pr[y|x, \pi]p(\pi|x)d\pi$ decomposes as

$$Var[y = k|x] = \underbrace{Var_{\pi \sim p(\pi|x)}[f_{true}^{\pi,k}(x)]}_{\text{Irreducible model uncertainty}} + \underbrace{\mathbb{E}_{\pi \sim p(\pi|x)}\left[(1 - f_{true}^{\pi,k}(x))f_{true}^{\pi,k}(x)\right]}_{\text{Data uncertainty}}.$$

The main source of uncertainty $p(\pi|x)$ is not a posterior this time but an empirical prior on a single observation $x$, hence its variance will not shrink as $|\mathcal{D}_{tr}| \to +\infty$. Therefore, the first term on the r.h.s. reflects the *irreducible model uncertainty* caused by the missing knowledge about $\pi$. The second term indicates data uncertainty as in PBMs, but accounts for the local uncertainty at $x$ caused by $\pi$. *Evidential Bayesian Models (EBMs)* (Sensoy et al., 2018; Malinin & Gales, 2018) suggest

$$p(y, \pi|x) = Pr[y|x, \pi]p(\pi|x) \approx Pr[y|\pi]q_\psi(\pi|x)$$

where $q_\psi(\pi|x)$ is a density function parameterized by $\psi$ and the dependency of the class-conditional on the input is dropped. Note that the resultant model employs uncertainty on individual data points via $\pi$, but it does not have any global random variables. The standard evidential model training is performed by maximizing the expected likelihood subject to a regularizer that penalizes the divergence of $q_\psi(\pi|x)$ from the prior belief

$$\arg\min_{\psi} - \int \log Pr[y|\pi]q_\psi(\pi|x)d\pi + \beta\text{KL}(q_\psi(\pi|x)||p(\pi)).$$

Although presented in the original work as as-hoc design choices, such a training objective can alternatively be viewed as $\beta$-regularized (Higgins et al., 2017) variational inference of $Pr[y|\pi]p(\pi|x)$ with the approximate posterior $q_\psi(\pi|x)$ (Chen et al., 2019).

## 4    COMPLETE BAYESIAN MODELS: BEST OF BOTH WORLDS

The uncertainty decomposition characteristics of PBMs and EBMs provide complementary strengths towards quantification of the three uncertainty types that constitute total calibration.

**i) Model misfit:**    The PBM is favorable since it provides shrinking posterior variance with growing training set size (Eq. 8) and the recovery of $\theta_{MLE}$ which inherits the consistency guarantees the frequentist approach. Since the uncertainty on the local variable $\pi$ does not shrink for large samples, NLL would measure how well $\psi$ can capture the regularities of heteroscedastic noise across individual samples, which is a more difficult task than quantifying the fit of a parametric model.

**ii) Class overlap:**    The EBM is favorable since its data uncertainty term $\mathbb{E}_{\pi \sim p(\pi|x_*)}[(1 - h^{\theta}_{\pi,k}(x_*))h^{\theta}_{\pi,k}(x_*)]$ is likely to have less estimator variance than $\mathbb{E}_{\theta \sim p(\theta|\mathcal{D}_{tr})}[(1 - h^{\theta}_{\pi,k}(x_*))h^{\theta}_{\pi,k}(x_*)]$ and calculating $p(\pi|x_*)$ does not require approximate inference as for $p(\theta|\mathcal{D}_{tr})$.

**iii) Domain mismatch:**    The PBM is favorable since the effect of the modulation of a global $\theta$ on the variance of $Z$ applies also to the posterior predictive distribution with the only difference that $x_*$ is a test sample. When the test sample is away from the training set, i.e. $\min_{x \in \mathcal{D}_{tr}} ||x_* - x||$ is large, then $p(y_*|x_*, \theta)$ is less likely to superpose with those regions of $\theta$ where $L(\theta)$ is pronounced, hence will be flattened out leading to a higher variance posterior predictive for samples coming from another domain. Since EBM has only local random variables, the variance of its posterior is less likely to build a similar discriminative property for domain detection.

> **Main Hypothesis.** *A model that inherits the model uncertainty of PBM and data uncertainty of EBM can simultaneously quantify: i) model misfit, ii) class overlap, and iii) domain mismatch.*

Guided by our main hypothesis, we combine the PBM and EBM designs within a unified framework that inherits the desirable uncertainty quantification properties of each individual approach

$$Pr[y|\pi]p(\pi|\theta, x)p(\theta), \tag{10}$$

which we refer to as a *Complete Bayesian Model (CBM)* hinting to its capability to solve the total calibration problem. The model simply introduces a global random variable $\theta$ into the empirical prior $p(\pi|\theta, x)$ of the EBM. The variance of the posterior predictive of the resultant model decomposes as

$$Var[y|x] = \underbrace{Var_{p(\theta|\mathcal{D})}\Big[\mathbb{E}_{p(\pi|x,\theta)}[\mathbb{E}[y|\pi, x]]\Big]}_{\text{Reducible Model Uncertainty}} + \underbrace{\mathbb{E}_{p(\theta|\mathcal{D})}\Big[Var_{p(\pi|\theta,x)}[\mathbb{E}[y|\pi]]\Big]}_{\text{Irreducible Model Uncertainty}}$$

$$+ \underbrace{\mathbb{E}_{p(\theta|\mathcal{D})}\Big[\mathbb{E}_{p(\pi|x,\theta)}[Var[y|\pi]]\Big]}_{\text{Data Uncertainty}} \tag{11}$$

where the r.h.s. of the decomposition has the following desirable properties: i) The first term recovers the form of the reducible model uncertainty term of PBM when $\pi$ is marginalized $Var_{p(\theta|\mathcal{D})}[\mathbb{E}_{p(\pi|x,\theta)}[\mathbb{E}[y|\pi, x]]] = Var_{p(\theta|\mathcal{D})}[\mathbb{E}[y|\theta, x]]$; ii) The second term is the irreducible model uncertainty term of EBM averaged over the uncertainty on the newly introduced global variable $\theta$. It quantifies the portion of uncertainty stemming from heteroscedastic noise, which is not explicit in the decomposition of PBM; iii) The third term recovers the data uncertainty term of EBM when $\theta$ is marginalized: $\mathbb{E}_{p(\theta|\mathcal{D})}[\mathbb{E}_{p(\pi|x,\theta)}[Var[y|\pi]]] = \mathbb{E}_{p(\pi|x)}[Var[y|\pi]]$.

## 5    THE EVIDENTIAL TURING PROCESS: AN EFFECTIVE CBM REALIZATION

Equipping the empirical prior $p(\pi|\theta, x)$ of CBM with the capability of *learning* to generate accurate prior beliefs on individual samples is the key for total calibration. We adopt the following two guiding principles to obtain highly expressive empirical priors: i) The existence of a global random variable $\theta$ can be exploited to express complex reducible model uncertainties using the Neural Processes (Garnelo et al., 2018b) framework, ii) The conditioning of $p(\pi|\theta, x)$ on a global variable $\theta$ and an individual observation $x$ can be exploited with an attention mechanism where $\theta$ is a memory and $x$ is a query. Dependency on a context set during test time can be lifted using the Neural Turing Machine

(Graves et al., 2014) design, which maintains an external memory that is updated by a rule detached from the optimization scheme of the main loss function. We extend the stochastic process derivation of the Neural Processes by marginalizing out a local variable from a PBM with an external memory that follows the Neural Turing Machine design and arrive at a new family of stochastic processes called a *Turing Process,* formally defined as below.

**Definition 1.** *A Turing Process is a collection of random variables $Y = \{y_i | i \in \mathcal{I}\}$ for an index set $\mathcal{I} = \{1, \ldots, K\}$ with arbitrary $K \in \mathbb{N}^+$ that satisfy the two properties*

$$(i) \ p_M(Y) = \int \prod_{y_i \in Y} p(y_i|\theta)p_M(\theta)d\theta, \quad (ii) \ p_{M'}(Y|Y') = \int \prod_{y_i \in Y} p(y_i|\theta)p_{M'}(\theta|Y')d\theta,$$

*for another random variable collection $Y'$ of arbitrary size that lives in the same probability space as $Y$, a probability measure $p_M(\theta)$ with free parameters $M$, and some function $M' = r(M, Y')$.*

The Turing Process can express all stochastic processes since property (i) is sufficient to satisfy Kolmogorov's Extension Theorem (Øksendal, 1992) and choosing $r$ simply to be an identity mapping would revert us back to the generic definition of a stochastic process. The Turing Process goes further by permitting to explicitly specify how information accumulates within the prior. This is a different approach from the Neural Process that performs conditioning $p(Y|Y')$ by applying an amortized posterior $q(\theta|Y')$ on a plain stochastic process that satisfies only Property (i), hence maintains a data-agnostic prior $p(\theta)$ and depends on a context set also at the prediction time.

Given a disjoint partitioning of the data $\mathcal{D} = \mathcal{D}_C \cup \mathcal{D}_T$ into a context set $(x', y') \in \mathcal{D}_C$ and a target set $(x, y) \in \mathcal{D}_T$, we propose the data generating process on the target set below

$$p(y, \mathcal{D}_T, \pi, \theta) = p(w) \underbrace{p_M(Z)}_{\substack{\text{External} \\ \text{memory}}} \prod_{(x,y) \in \mathcal{D}_T} \Big[ p(y|\pi) \underbrace{p(\pi|Z, w, x)}_{\substack{\text{Input-specific} \\ \text{prior}}} \Big]. \tag{12}$$

We decouple the global parameters $\theta = \{w, Z\}$ into a $w$ that parameterizes a map from the input to the hyperparameter space and an external memory $Z = \{z_1, \ldots, z_R\}$ governed by the distribution $p_M(Z)$ parameterized by $M$ that embeds the context data during training. The memory variable $Z$ updates its belief conditioned on the context set $\mathcal{D}_C$ by updating its parameters with an explicit rule $p_M(Z|\mathcal{D}_C) \leftarrow p_{\mathbb{E}_{Z \sim p_M(Z)}[r(Z,\mathcal{D}_C)]}(Z)$. The hyperpriors of $p(\pi|Z, w, x)$ are then determined by querying the memory $Z$ for each input observation $x$ using an attention mechanism, for instance a transformer network (Vaswani et al., 2017). Choosing the variational distribution as $q_\lambda(\theta, \pi|x) = q_\lambda(w)p_M(Z)q(\pi|Z, w, x)$, we attain the variational free energy formula for our target model

$$\mathcal{F}_{ETP}(\lambda) = \tag{13}$$

$$\mathbb{E}_{q_\lambda(w)} \left[ -\sum_{(x,y) \in \mathcal{D}} \mathbb{E}_{q_\lambda(\pi|Z,w,x)} \left[ \mathbb{E}_{p_M(Z)} \left[ \log p(y|\pi) + \log \frac{q_\lambda(\pi|Z, w, x)}{p(\pi|Z, w, x)} \right] \right] + \log \frac{q_\lambda(w)}{p(w)} \right].$$

The learning procedure will then alternate between updating the variational parameters $\lambda$ via gradient-descent on $\mathcal{F}_{ETP}(\lambda)$ and updating the parameters of the memory variable $Z$ via an external update rule as summarized in Figure 2b. ETP can predict a test sample $(x_*, y_*)$ by only querying the input $x_*$ on the memory $Z$ and evaluating the prior $p(\pi_*|Z, w, x_*)$ without need for any context set as

$$p(y_*|\mathcal{D}_{tr}, x_*) \approx \iint p(y_*|\pi_*)q(\pi_*|Z, w, x_*)p_M(Z)q_\lambda(w)dZdw. \tag{14}$$

## 6 MOST IMPORTANT SPECIAL CASES OF THE EVIDENTIAL TURING PROCESS

As we build the design of the Evidential Turing Process (ETP) on the CBM framework that over-arches the prevalent parametric and evidential approaches, its ablation amounts to recovering many state-of-the-art modeling approaches as listed in Table 1 and detailed below.

**Bayesian Neural Net (BNN)** (Neal, 1995; MacKay, 1992; 1995) is the most characteristic PBM example that parameterize a likelihood $p(y|\theta, x)$ with a neural network $f_\theta(\cdot)$ whose weights follow a distribution $\theta \sim p(\theta)$. In the experiments we assume as a prior $\theta \sim \mathcal{N}(\theta|0, \beta^{-1}I)$ and infer its posterior by mean-field variational posterior $q_\lambda(\theta)$ applying the reparameterization trick (Kingma et al., 2015; Molchanov et al., 2017).

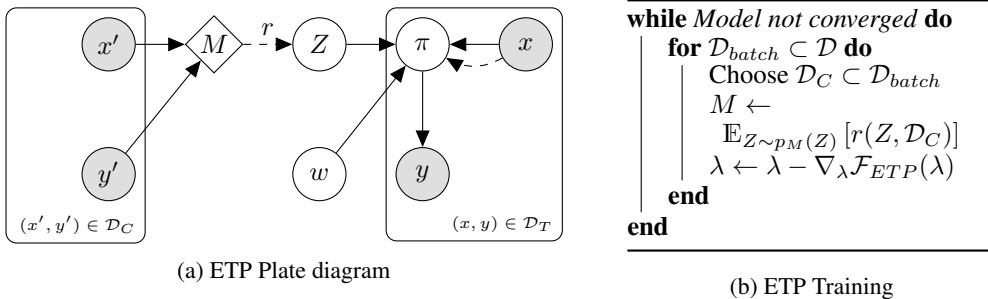

(a) ETP Plate diagram

(b) ETP Training

Figure 2: The Evidential Turing Process: *(left)* The essential design choice is how $p(\pi|Z, w, x)$ is constructed thanks to an external memory $M$ that can be queried with respect to any input $x$ without needing to store context data at test time. The dashed arrow indicates that $Z$ can condition on a context $\mathcal{D}_C$ by updates its parameters $M$ with an explicit function $r$. *(right)* The ETP training routine.

Table 1: **Ablation Table.** Deactivating the components of ETP one at a time equates it to state-of-the-art methods. We curate ENP as a surrogate for the Attentive NP (ANP) of (Kim et al., 2019), which requires test-time context, and an improved NP variant with reduced estimator variance thanks to $\pi$.

| Model | $\pi$ | $Z$ | $M$ | $r$ | $\mathcal{D}_c^{(test)}$ | Compare |
|---|---|---|---|---|---|---|
| ETP (Target) | ✓ | ✓ | ✓ | ✓ | ✗ | |
| ANP (Kim et al., 2019) | ✓ | ✓ | ✓ | ✗ | ✓ | No |
| ENP (Surrogate) | ✓ | ✓ | ✗ | ✗ | ✗ | Yes |
| NP (Garnelo et al., 2018b) | ✗ | ✓ | ✗ | ✗ | ✗ | No |
| EDL (Sensoy et al., 2018) | ✓ | ✗ | ✗ | ✗ | ✗ | Yes |
| BNN (Molchanov et al., 2017) | ✗ | ✗ | ✗ | ✗ | ✗ | Yes |

✓: Component active      ✗: Component inactive
$\mathcal{D}_c^{(test)}$: Demand for context data at test time

**Evidential Deep Learning (EDL)** (Sensoy et al., 2018) introduces the characteristic example of EBMs. EDL employs an uninformative Dirichlet prior on the class probabilities, which then set the mean of a normal distribution on the one-hot-coded class labels $y, \pi \sim \mathcal{D}ir(\pi|1, \ldots, 1)\mathcal{N}(y|\pi, 0.5I_K)$. EDL links the input observations to the distribution of class probabilities by performing amortized variational inference with the approximate distribution $q_\lambda(\pi; x) = \mathcal{D}ir(\pi|\alpha_\lambda(x))$.

**Neural Processes (NP)** (Garnelo et al., 2018a;b) is a PBM used to generate stochastic processes from neural networks by integrating out the global parameters $\theta$ as in Property (i) of Definition 1. NPs differ from PBMs by amortizing an arbitrary sized context set $\mathcal{D}_C$ on the global parameters $q(\theta|\mathcal{D}_C)$ via an aggregator network during inference. NPs can be viewed as Turing Processes with an identity map $r$. Follow-up work equipped NPs with attention (Kim et al., 2019), translation equivariance (Gordon et al., 2020; Foong et al., 2020), and sequential prediction (Singh et al., 2019; Yoon et al., 2020). All of these NP variants assume prediction-time access to context, making them suitable for interpolation tasks such as image in-painting but not for generic prediction problems.

**Evidential Neural Process (ENP)** is the EBM variant of a neural process we introduce to bring together best practices of EDL and NPs and make the strongest possible baseline for ETP, defined as

$$p(y, \pi, Z|\mathcal{D}_C) = \mathcal{C}at(y|\pi)\mathcal{D}ir\big(\pi|\alpha_\theta(Z)\big)\mathcal{N}\big(Z|(\mu, \sigma^2) = r(\{h_\theta(x_j, y_j) \mid (x_j, y_j) \in \mathcal{D}_C\})\big), \quad (15)$$

with an aggregation rule $r(\cdot)$, an encoder net $h_\theta(\cdot, \cdot)$, and a neural net $\alpha_\theta(\cdot)$ mapping the global variable $Z$ to the class probability simplex. We further improve upon ENP with an attentive NP approach (Kim et al., 2019) by replacing the fixed aggregation rule $r$ with an attention mechanism, thereby allowing the model to switch focus depending on the target. At the prediction time, we use $Z \sim \mathcal{N}(Z|\mathbf{1}, \kappa^2 I)$ for small $\kappa$ to induce an uninformative prior on $\pi$ in the absence of a context set.

## 7 CASE STUDY: CLASSIFICATION WITH EVIDENTIAL TURING PROCESSES

We demonstrate a realization of an Evidential Turing Process to a classification problem below

$$y, \pi, w, Z | M \sim \mathcal{C}at(y|\pi)\mathcal{D}ir\big(\pi| \exp(a(v_w(x); Z))\big)\mathcal{N}(w|0, \beta^{-1}I) \prod_{r=1}^{R} \mathcal{N}(z_r|m_r, \kappa^2 I). \quad (16)$$

The global variable $Z$ is parameterized by the memory $M = (m_1, \ldots, m_R)$ consisting of $R$ cells $m_r \in \mathbb{R}^K$.[1] The input data $x$ is mapped to the same space via an encoding neural net $v_w(\cdot)$ parameterizing a Dirichlet distribution over the class probabilities $\pi$ via an attention mechanism $a(v_w, z) = \sum_{z' \in Z} \phi(v_w(x), z')z$, where $\phi(v_w(x), z) = \text{softmax}(\{k_\psi(z)^\top v_w(x)/\sqrt{K} | z \in Z\})$. The function $k_\psi(\cdot)$ is the key generator network operating on the embedding space. We update the memory cells as a weighted average between remembering and updating

$$m \leftarrow \mathbb{E}_{Z \sim p(Z|M)} \left[ \tanh\left(\gamma m + (1-\gamma) \sum_{(x,y) \in \mathcal{D}_C} \phi(v_w(x), z)[\text{onehot}(y) + \text{soft}(v_w(x))]\right)\right] \quad (17)$$

where $\gamma \in (0, 1)$ is a fixed scaling factor controlling the relative importance. The second term adds new information to the cell as a weighted sum over all pairs $(x, y)$, taking both the true label (via onehot$(y)$) as well as the uncertainty in the prediction (via soft$(v_w(x))$) into account. The final $\tanh(\cdot)$ transformation ensures that the memory content remains in a fixed range across updates, as practised in LSTMs (Hochreiter & Schmidhuber, 1997).

## 8 EXPERIMENTS

We benchmark ETP against the state of the art as addressed in the ablation plan in Table 1 according to the total calibration criteria developed in Sec. 2 on five real-world data sets. See the Appendix B for full details on the experimental setup in each case and for the hyperparameters used throughout. We provide a reference implementation of the proposed model and the experimental pipeline.[2]

**Total calibration.** Table 2 reports the prediction error and the three performance scores introduced in Sec. 2 that constitute total calibration. We perform the out-of-domain detection task as in Malinin & Gales (2018) and classify the test split of the target domain and a data set from another domain based on the predictive entropy. ETP is the only method that consistently ranks among top performers in all data sets with respect to all performance scores, which supports our main hypothesis that CBM should outperform PBM and EBM in total calibration.

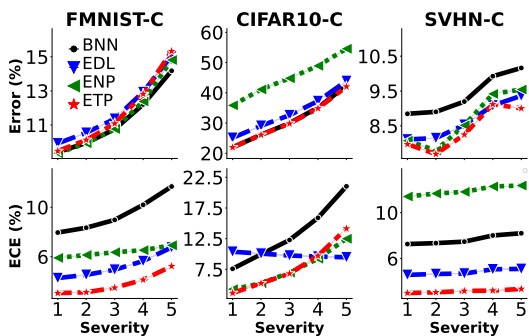

Figure 3: Results averaged across 19 types of corruption (e.g. motion blur, fog, pixelation) applied on the test splits of FMNIST, CIFAR10, and SVHN at five severity levels.

**Response to gradual domain shift.** In order to assess how well models can cope with a gradual transition from their native domain, we evaluate their ECE performance on data perturbed by 19 types of corruption (Hendrycks & Dietterich, 2019) at five different severity levels. Figure 3 depicts the performance of models averaged across all corruptions. ETP gives the best performance nearly in all data sets and distortion severity levels (see Appendix B.3 Table 4 for a tabular version of these results).

**Computational cost.** We measured the average wall-clock time per epoch in CIFAR10 to be $10.8 \pm 0.1$ seconds for ETP, $8.2 \pm 0.1$ seconds for BNN, $8.6 \pm 0.1$ seconds for EDL, and $9.5 \pm 0.2$ seconds for ENP. The relative standings of the models remain unchanged in all the other data sets.

---

[1]In the experiments we fix $\kappa^2 = 0.1$. Preliminary results showed stability as well for larger/smaller values.
[2]https://github.com/ituvisionlab/EvidentialTuringProcess

Table 2: Quantitative results on five data sets showing mean $\pm$ standard deviation across 10 repetitions. Best performing models that overlap within three standard deviations are highlighted in bold.

| Domain Data (Architecture) | IMDB (LSTM) | Fashion (LeNet5) | SVHN (LeNet5) | CIFAR10 (LeNet5) | CIFAR100 (ResNet18) |
|---|---|---|---|---|---|
| Prediction accuracy as % test error | | | | | |
| BNN | $16.4 \pm 0.6$ | $7.9 \pm 0.1$ | $7.9 \pm 0.1$ | $15.3 \pm 0.3$ | $30.2 \pm 0.3$ |
| EDL | $38.3 \pm 13.3$ | $8.6 \pm 0.1$ | $7.3 \pm 0.1$ | $18.5 \pm 0.2$ | $45.2 \pm 0.4$ |
| ENP | $50.0 \pm 0.0$ | $7.9 \pm 0.2$ | $6.7 \pm 0.1$ | $14.8 \pm 0.2$ | $39.0 \pm 0.3$ |
| ETP (Target) | $15.8 \pm 1.3$ | $7.9 \pm 0.2$ | $6.9 \pm 0.1$ | $15.3 \pm 0.2$ | $29.2 \pm 0.3$ |
| In-domain calibration as % Expected Calibration Error (ECE) | | | | | |
| BNN | $14.4 \pm 0.4$ | $6.7 \pm 0.$ | $6.5 \pm 0.$ | $5.5 \pm 0.3$ | $15.2 \pm 0.0$ |
| EDL | $41.1 \pm 2.6$ | $3.7 \pm 0.2$ | $4.0 \pm 0.1$ | $9.0 \pm 0.2$ | $5.3 \pm 0.4$ |
| ENP | $0.8 \pm 1.6$ | $6.0 \pm 0.2$ | $10.7 \pm 0.2$ | $7.2 \pm 0.3$ | $39.7 \pm 0.4$ |
| ETP (Target) | $3.1 \pm 0.4$ | $2.6 \pm 0.2$ | $2.6 \pm 0.1$ | $2.7 \pm 0.1$ | $6.6 \pm 0.1$ |
| Model fit as negative test log-likelihood | | | | | |
| BNN | $0.47 \pm 0.0$ | $0.65 \pm 0.0$ | $0.71 \pm 0.0$ | $0.50 \pm 0.0$ | $1.78 \pm 0.0$ |
| EDL | $0.66 \pm 0.1$ | $0.37 \pm 0.0$ | $0.34 \pm 0.0$ | $0.72 \pm 0.0$ | $2.24 \pm 0.0$ |
| ENP | $0.69 \pm 0.0$ | $0.34 \pm 0.0$ | $0.33 \pm 0.0$ | $0.50 \pm 0.0$ | $2.52 \pm 0.0$ |
| ETP (Target) | $0.37 \pm 0.0$ | $0.29 \pm 0.0$ | $0.26 \pm 0.0$ | $0.46 \pm 0.0$ | $1.36 \pm 0.0$ |
| Out-of-domain detection as % Area Under ROC Curve | | | | | |
| OOD Data | Random | MNIST | CIFAR100 | SVHN | TinyImageNet |
| BNN | $60.9 \pm 4.2$ | $75.9 \pm 2.3$ | $86.2 \pm 0.5$ | $84.1 \pm 1.3$ | $97.2 \pm 0.5$ |
| EDL | $55.1 \pm 5.1$ | $77.5 \pm 2.0$ | $90.9 \pm 0.3$ | $79.2 \pm 0.7$ | $89.6 \pm 0.3$ |
| ENP | $53.7 \pm 5.7$ | $88.9 \pm 1.0$ | $92.4 \pm 0.4$ | $81.4 \pm 0.8$ | $100.0 \pm 0.1$ |
| ETP (Target) | $59.1 \pm 5.1$ | $90.0 \pm 0.9$ | $90.0 \pm 0.4$ | $82.1 \pm 0.6$ | $99.6 \pm 0.1$ |

## 9 CONCLUSION

**Summary.** We initially develop the first formal definition of total calibration. Then we analyze how the design of two mainstream Bayesian modeling approaches affect their uncertainty characteristics. Next we introduce Complete Bayesian Models as a unifying framework that inherits the complementary strengths existing ones. We develop the Evidential Turing Process as an optimal realization of Complete Bayesian Models. We derive an experiment setup from our formal definition of total calibration and a systematic ablation of our target model into the strongest representatives of the state of the art. We observe in five real-world tasks that the Evidential Turing Process is the only model that can excel all three aspects of total calibration simulatenously.

**Broad impact.** Our work delivers evidence for the claim that epistemic awareness of a Bayesian model is indeed a capability learnable only from in-domain data, as appears in biological intelligence via closed-loop interactions of neuronal systems with stimuli. The implications of our work could follow up in interdisciplinary venues, with focus on the relation of associative memory and attention to the neuroscientific roots of epistemic awareness (Lorenz, 1978).

**Limitations and ethical concerns.** While we observed ETP to outperform BNN variants in our experiments, whether BNNs could bridge the gap after further improvements in the approximate inference remains an open question. The attention mechanism of ETP could also be improved, for instance to transformer nets (Vaswani et al., 2017). The explainability of the memory content of ETP deserves further investigation. The generalizeability of our results to very deep architectures, such as ResNet 152 or DenseNet 161 could be a topic of a separate large-scale empirical study. ETP does not improve the explainability and fairness of the decisions made by the underlying design choices, such as the architecture of the used neural nets in the pipeline. Potential negative societal impacts of deep neural net classifiers stemming from these two factors need to be circumvented separately before the real-world deployment of our work.

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

# APPENDIX

## A FURTHER ANALYTICAL RESULTS AND DERIVATIONS

### A.1 KULLBACK LEIBLER BETWEEN TWO DIRICHLET DISTRIBUTIONS

As both our ETP and and the EDL baseline use the Kullback-Leibler divergence between two Dirichlet distributions, we give the full details of its calculation here. For two distributions over a $K$-dimensional probability $\pi$, $\mathcal{D}ir(\pi|\alpha)$ and $\mathcal{D}ir(\pi|\beta)$, parameterized by $\alpha, \beta \in \mathbb{R}_+^K$, the following identity holds

$$\text{KL}\left(\mathcal{D}ir(\pi|\alpha) \parallel \mathcal{D}ir(\pi|\beta)\right) = \log\left(\frac{\Gamma(\sum_k \alpha_k)\prod_k \Gamma(\beta_k)}{\Gamma(\sum_k \beta_k)\prod_k \Gamma(\alpha_k)}\right) + \sum_k (\alpha_k - \beta_k)\big(\psi(\alpha_k) - \psi(\textstyle\sum_k \alpha_k)\big),$$

where $\Gamma(\cdot)$ and $\psi(a) := \frac{d}{da}\log\Gamma(a)$ are the *gamma* and *digamma* functions (Abramowitz & Stegun, 1965).

### A.2 THE EVIDENTIAL DEEP LEARNING OBJECTIVE

The analytical expression of the Evidential Deep Learning (Sensoy et al., 2018) loss as defined in the main paper can be computed as follows. For a single $K$-dimensional observation $y$, dropping the $n$ from the notation and instead using the index for the dimensionality throughout the following equations, we have

$$
\begin{aligned}
\mathbb{E}_{p(\pi|x)}\left[||y - \pi||^2\right] &= \mathbb{E}_{p(\pi|x)}\left[(y-\pi)^\top(y-\pi)\right] \\
&= \sum_{k=1}^K \mathbb{E}_{p(\pi|x)}\left[(y_k - \pi_k)^2\right] \\
&= \sum_{k=1}^K \mathbb{E}_{p(\pi|x)}\left[(y_k - \mathbb{E}_{p(\pi|x)}\left[\pi_k\right] + \mathbb{E}_{p(\pi|x)}\left[\pi_k\right] - \pi_k)^2\right] \\
&= \sum_{k=1}^K (y_k - \mathbb{E}_{p(\pi|x)}\left[\pi_k\right])^2 + \text{var}_{p(\pi|x)}\left[\pi_k\right],
\end{aligned}
$$

with the tractable expectation and variance of a Dirichlet distributed $\pi$. The Kullback-Leibler term is between two Dirichlet distributions and given (see the general form in A.1) as

$$
\begin{aligned}
&\text{KL}\left(\mathcal{D}ir(\pi|\alpha_\theta(x)) \parallel \mathcal{D}ir(\pi|1,\dots,1)\right) \\
&= \log\left(\frac{\Gamma\left(\sum_k \alpha_\theta(x)_k\right)}{\Gamma(K)\prod_k \Gamma(\alpha_\theta(x)_k)}\right) + \sum_{k=1}^K (\alpha_\theta(x)_k - 1)\Big(\psi(\alpha_\theta(x)_k) - \psi\big(\textstyle\sum_k \alpha_\theta(x)_k\big)\Big).
\end{aligned}
$$

### A.3 EVIDENTIAL DEEP LEARNING AS A LATENT VARIABLE MODEL

As discussed in the main paper for a set of $N$ observations $\mathcal{D} = \{(x_1, y_1), \dots, (x_N, y_N)\}$, the EDL objective minimizes is given as

$$\mathcal{L}_{\text{EDL}} = \sum_{n=1}^N \mathbb{E}_{p(\pi_n|x_n)}\left[||y_n - \pi_n||_2^2\right] + \lambda\text{KL}\left(p(\pi_n|x_n) \parallel \mathcal{D}ir(\pi_n|1,\dots,1)\right),$$

where $p(\pi_n|x_n) = \mathcal{D}ir(\pi|\alpha_\theta(x_n))$. We can instead assume the following generative model

$$
\begin{aligned}
\pi_n &\sim \mathcal{D}ir(\pi_n|1,\dots,1) &&\forall n \\
y_n &\sim \mathcal{N}(y_n|\pi_n, 0.5 I_K) &&\forall n,
\end{aligned}
$$

i.e. latent variables $\pi_n$ and $K$-dimensional observations $y_n$ following a multivariate normal prior. Approximating the intractable posterior $p(\boldsymbol{\pi}|\mathbf{y})$, where $\boldsymbol{\pi} = (\pi_1, \dots, \pi_n)$, $\mathbf{y} = (y_1, \dots, y_n)$, with an

amortized variational posterior $q(\pi_n; x_n) = \mathcal{D}ir(\pi_n|\alpha_\theta(x_n))$, where $\alpha_\theta(\cdot)$ is the same architecture as in the EDL model, we have as the *evidence lower bound (ELBO)* to be maximized

$$\sum_{n=1}^{N} \mathbb{E}_{q(\pi_n; x_n)} \left[\log p(y_n|\pi_n)\right] - \mathrm{KL}\left(q(\pi_n; x_n) \parallel p(\pi_n)\right)$$

$$= \sum_{n=1}^{N} \mathbb{E}_{q(\pi_n; x_n)} \left[-\frac{1}{2}\log(2\pi) + \frac{K}{2}\log(2) - (y_n - \pi_n)^\top (y_n - \pi_n)\right] - \mathrm{KL}\left(q(\pi_n; x_n) \parallel p(\pi_n)\right)$$

$$= \mathrm{const} - \sum_{n=1}^{N} \mathbb{E}_{q(\pi_n; x_n)} \left[(y_n - \pi_n)^\top (y_n - \pi_n)\right] - \mathrm{KL}\left(q(\pi_n; x_n) \parallel p(\pi_n)\right)$$

$$= \mathrm{const} - \mathcal{L}_{\mathrm{EDL}},$$

that is the negative ELBO is equal to the EDL loss up to an additive constant.

### A.4 POSTERIOR PREDICTIVE

For a new input observation $x^*$, the corresponding posterior predictive distribution on the output $y^*$ can be computed as

$$p(y^* = k|x^*, \mathcal{D}) \approx \mathbb{E}_{p(Z|M)} \left[\mathbb{E}_{q_\lambda(\pi|x^*, Z)} \left[p(y^* = k|\pi)\right]\right] = \mathbb{E}_{p(Z|M)} \left[h_\theta^k / \sum_{k'} h_\theta^{k'}\right],$$

where $h_\theta^k = h_\theta\left(v_\theta(x^*), a(v_\theta(x^*); Z)\right)_k$. That is, we can compute it analytically up to a sampling-based evaluation of the expectation over $p(Z|M)$.

## B EXPERIMENTAL DETAILS

### B.1 SYNTHETIC 1D TWO-CLASS CLASSIFICATION

We illustrate the qualitative behaviour of ETP on a synthetic 1d two-class classification task. The data consist of observations (20 per class) from two highly overlapping Gaussian distributions. The neural net, a simple one hidden layer architecture, can learn the task with high accuracy. Figure 4 shows the raw data and underlying distributions as well as the learned memory evidence. The model learns to place more evidence on the correct class further away from the decision boundary while also becoming more varied with increasing distance. Within the high-density region, the asymmetry in how the specific observations are spread out leads to an asymmetry in the memory evidence. Below zero, the blue points are clearly separated, leading to a quick emphasis on that class as we move towards the left. Above zero, however, some blue observations in the orange region prevent the model from rapidly developing over-confidence.

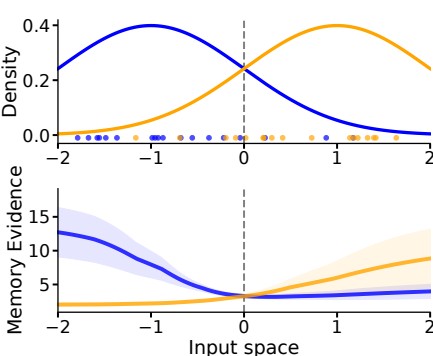

Figure 4: **1D Classification Task.** The upper plot shows the underlying distributions of each of the two classes, as well as the observed data. The lower shows the regularizing evidence the generative model places on each of the two classes depending on location in space, that is, mean $\pm$ one standard deviation over ten samples from $p(Z|M)$.

This synthetic data set consisting of two classes with 20 observations each sampled from standard normal distributions centered at $\pm 1$ respectively. The encoding function $v_\phi(\cdot)$ consists of a multi-layer perceptron with a single hidden layer consisting of 32 neurons and a ReLU activation function. It is optimized for 400 epochs with the Adam optimizer (Kingma & Ba, 2015) using the PyTorch (Paszke et al., 2019) default parameters and a learning rate of 0.001. In order to keep the model as simple

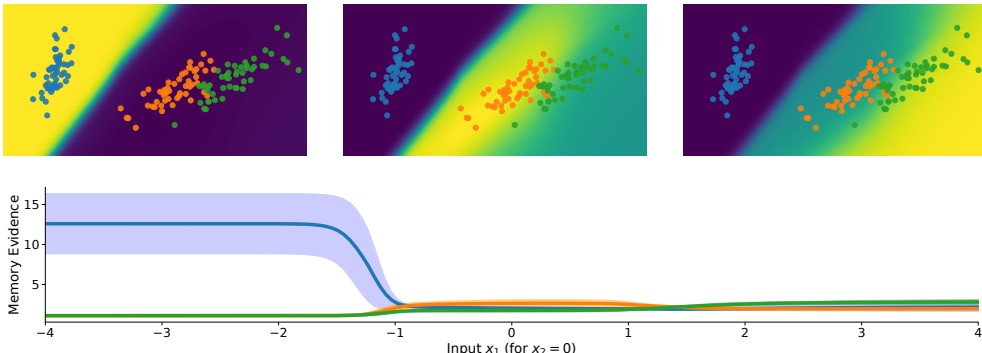

Figure 5: **2D Classification Task.** The three upper plots visualize the regularizing evidence that the model places on the location in space as the average over ten samples from $p(Z|M)$. The color scales are different across the three plots and vary in overall intensity. The lower plot visualizes a horizontal cut through the middle of these plots, putting them on a common scale. The solid lines in each color indicate the mean memory evidence, and the corresponding shaded areas indicate one standard deviation. Around the separable blue class, the memory strongly emphasizes the correct class with large confidence and suppresses the other two classes depicted in orange and green. While always preferring the correct class, the memory regularizes against overconfidence around the overlapping orange and green classes.

as possible, the key generator function $k_\lambda(\cdot)$ is constrained to being the identity function, while $h_\xi\big(v(x), a(v(x); Z)\big) = v(x) + \tanh(a(v(x); Z)$. The memory update is simplified to dropping the $\tanh(\cdot)$ rescaling.

## B.2 IRIS 2D THREE-CLASS CLASSIFICATION

The data set used in this experiment is the classical Iris data set[3] consisting of 150 samples of three iris species (50 per class), with four features measured per sample. We first map the data to the first two principal components for visualization and also use this modified version as the training data. This gives an interpretable toy learning setup where one class is clearly separated from the other two overlapping classes. The encoding function $v_\phi(\cdot)$ consists of an MLP with two hidden layers of 32 neurons each and ReLU activation functions. We train it for 400 epochs with the Adam optimizer using the PyTorch default parameters and a learning rate of 0.001. In order to keep the model as simple as possible, we constrain the key generator function $k_\lambda(\cdot)$ to the identity function while setting $h_\xi\big(v(x), a(v(x); Z)\big) = v(x) + \tanh(a(v(x); Z)$. We simplify the memory update by dropping the $\tanh(\cdot)$ rescaling. Figure 5 shows the varying importance the model assigns to different parts of the input space depending on the class under consideration. For the separate blue class, the model is significantly more confident in relative terms and absolute values.

## B.3 EXPERIMENTS ON REAL DATA

All experiments are implemented in PyTorch (Paszke et al., 2019) version 1.7.1 and trained on a TITAN RTX. They have been replicated ten times over random initial seeds.

**Fashion MNIST (FMNIST).** We use a LeNet5-sized architecture (see Table 3). The out-of-distribution data is the MNIST (LeCun et al., 2010) data set. We z-score normalize all data sets with the in-domain mean and standard deviations. We train each model for 50 epochs with the Adam optimizer (Kingma & Ba, 2015) using a learning rate of 0.001.

---

[3]Originally due to Fisher (1936) and Anderson (1935). We rely on the version provided by the scikit learn library, see https://scikit-learn.org/stable/modules/classes.html#module-sklearn.datasets.

Table 3: The neural network architectures used for the FMNIST, CIFAR10, and SVHN data set with ReLU activations between the layers.

| FMNIST | CIFAR10/SVHN |
|---|---|
| Convolution ($5 \times 5$) with 20 channels | Convolution ($5 \times 5$) with 192 channels |
| MaxPooling ($2 \times 2$) with stride 2 | |
| Convolution ($5 \times 5$) with 50 channels | Convolution ($5 \times 5$) with 192 channels |
| MaxPooling ($2 \times 2$) with stride 2 | |
| Linear with 500 neurons | Linear with 1000 neurons |
| Linear with 10 neurons | |

| CIFAR100 | |
|---|---|
| Layer Name | ResNet-18 |
| Conv1 | $7 \times 7, 64$, stride 2 |
| | $3 \times 3$ max pool, stride 2 |
| Conv2_x | $\begin{bmatrix} 3 \times 3, 64 \\ 3 \times 3, 64 \end{bmatrix} \times 2$ |
| Conv3_x | $\begin{bmatrix} 3 \times 3, 128 \\ 3 \times 3, 128 \end{bmatrix} \times 2$ |
| Conv4_x | $\begin{bmatrix} 3 \times 3, 256 \\ 3 \times 3, 256 \end{bmatrix} \times 2$ |
| Conv5_x | $\begin{bmatrix} 3 \times 3, 512 \\ 3 \times 3, 512 \end{bmatrix} \times 2$ |
| Average Pool | $7 \times 7$ average pool |
| Linear | 100 neurons |

| IMDB | |
|---|---|
| Layer Name | LSTM |
| Embedding | $1001 \rightarrow 64$ |
| LSTM | 2 layers with 256 hidden size |
| Linear | 2 neurons |

**CIFAR10 (C10).** We use a LeNet5-sized architecture (see Table 3). The out-of-distribution data is the SVHN (Netzer et al., 2011). We z-score normalize all data sets with the in-domain mean and standard deviations. We further rely on data augmentation for the in domain data using random crops with 32 pixels and four pixel padding as well as horizontal flips. We train each model for 100 epochs using the Adam optimizer (Kingma & Ba, 2015) with a learning rate of 0.0001.

**SVHN.** We use a LeNet5-sized architecture (see Table 3). The out-of-distribution data is the CIFAR10 data set. We z-score normalize all data sets with the in-domain mean and standard deviations. We train each model for 100 epochs using the Adam optimizer (Kingma & Ba, 2015) with a learning rate of 0.0001.

**IMDB Sentiment Classification.** We use a LSTM architecture with 64 embedding dimensions and 2 layers with 256 hidden dimensions. We generate the out-of-distribution data by sampling values from the $[1, 1000]$ interval uniformly at random. We train each model for 20 epochs using the SGD optimizer (Robbins & Monro, 1951) with a learning rate of 0.05 and 0.9 momentum. For data preparation, we follow the source code of the original work [4].

**Shared Details and Observations.** In all experiments, we train the BNN models with a cross-entropy loss using softmax as the squashing function to calculate class probabilities. The EDL model is trained with the original loss (Sensoy et al., 2018). We make our predictions with the mean of the Dirichlet distribution as in the original work. For all models, we use entropy as the criterion for detecting out-of-distribution instances.

---

[4] https://www.kaggle.com/arunmohan003/sentiment-analysis-using-lstm-pytorch

Table 4: Quantitative results of models on the FMNIST-C, CIFAR10-C and SVHN-C test dataset using the LeNet5. The table below reports mean $\pm$ three standard deviations.

| Severity | Method | FMNIST-C | | CIFAR10-C | | SVHN-C | |
|---|---|---|---|---|---|---|---|
| | | Err (%) ($\downarrow$) | ECE (%) ($\downarrow$) | ($\downarrow$) | ECE (%) ($\downarrow$) | ($\downarrow$) | ECE (%) ($\downarrow$) |
| 1 | BNN | $9.4_{\pm2.8}$ | $8.0_{\pm2.4}$ | $21.9_{\pm4.5}$ | $7.6_{\pm2.9}$ | $8.8_{\pm1.2}$ | $7.3_{\pm1.0}$ |
| | EDL | $9.9_{\pm2.6}$ | $4.3_{\pm1.0}$ | $25.1_{\pm4.6}$ | $10.4_{\pm1.3}$ | $8.1_{\pm1.2}$ | $4.6_{\pm0.6}$ |
| | ENP | $9.3_{\pm2.7}$ | $5.9_{\pm0.5}$ | $35.9_{\pm6.0}$ | $4.3_{\pm1.7}$ | $8.1_{\pm1.3}$ | $11.4_{\pm1.0}$ |
| | ETP | $9.5_{\pm2.8}$ | $3.0_{\pm1.0}$ | $21.9_{\pm4.6}$ | $3.6_{\pm2.2}$ | $8.0_{\pm1.2}$ | $3.0_{\pm0.4}$ |
| 2 | BNN | $9.9_{\pm2.3}$ | $8.3_{\pm1.9}$ | $26.1_{\pm5.3}$ | $10.0_{\pm3.7}$ | $8.9_{\pm1.2}$ | $7.3_{\pm0.9}$ |
| | EDL | $10.5_{\pm2.1}$ | $4.5_{\pm0.8}$ | $29.1_{\pm5.1}$ | $10.1_{\pm1.6}$ | $8.2_{\pm1.1}$ | $4.6_{\pm0.6}$ |
| | ENP | $9.9_{\pm2.3}$ | $6.1_{\pm1.0}$ | $41.1_{\pm6.6}$ | $5.2_{\pm2.1}$ | $7.8_{\pm1.2}$ | $11.7_{\pm1.1}$ |
| | ETP | $10.1_{\pm2.4}$ | $3.1_{\pm0.8}$ | $26.1_{\pm5.3}$ | $5.2_{\pm2.8}$ | $7.7_{\pm1.1}$ | $3.0_{\pm0.5}$ |
| 3 | BNN | $10.7_{\pm2.3}$ | $9.0_{\pm1.9}$ | $30.0_{\pm7.6}$ | $12.3_{\pm5.3}$ | $9.2_{\pm1.6}$ | $7.5_{\pm1.2}$ |
| | EDL | $11.4_{\pm2.1}$ | $4.9_{\pm0.9}$ | $32.6_{\pm7.0}$ | $9.8_{\pm1.5}$ | $8.5_{\pm1.5}$ | $4.7_{\pm0.7}$ |
| | ENP | $10.8_{\pm2.3}$ | $6.4_{\pm1.6}$ | $44.8_{\pm9.0}$ | $7.0_{\pm3.4}$ | $8.5_{\pm1.7}$ | $11.8_{\pm1.5}$ |
| | ETP | $11.1_{\pm2.4}$ | $3.5_{\pm0.9}$ | $29.8_{\pm7.2}$ | $6.8_{\pm3.8}$ | $8.2_{\pm1.5}$ | $3.2_{\pm0.6}$ |
| 4 | BNN | $12.2_{\pm4.2}$ | $10.2_{\pm3.6}$ | $35.1_{\pm10.9}$ | $15.9_{\pm8.3}$ | $9.9_{\pm2.5}$ | $8.0_{\pm1.9}$ |
| | EDL | $12.9_{\pm4.2}$ | $5.6_{\pm1.8}$ | $37.3_{\pm10.0}$ | $9.6_{\pm1.4}$ | $9.1_{\pm2.4}$ | $5.1_{\pm1.0}$ |
| | ENP | $12.4_{\pm4.3}$ | $6.5_{\pm2.2}$ | $48.9_{\pm11.2}$ | $9.2_{\pm5.4}$ | $9.4_{\pm2.8}$ | $12.3_{\pm2.2}$ |
| | ETP | $12.8_{\pm4.6}$ | $4.1_{\pm1.8}$ | $34.8_{\pm10.3}$ | $9.7_{\pm6.4}$ | $9.1_{\pm2.4}$ | $3.2_{\pm1.0}$ |
| 5 | BNN | $14.2_{\pm5.4}$ | $11.7_{\pm4.6}$ | $42.4_{\pm13.7}$ | $21.0_{\pm10.4}$ | $10.2_{\pm2.5}$ | $8.2_{\pm1.9}$ |
| | EDL | $15.1_{\pm5.8}$ | $6.7_{\pm3.1}$ | $43.9_{\pm12.5}$ | $9.5_{\pm2.4}$ | $9.4_{\pm2.4}$ | $5.1_{\pm1.1}$ |
| | ENP | $14.8_{\pm6.5}$ | $6.9_{\pm3.1}$ | $54.6_{\pm12.7}$ | $12.5_{\pm7.0}$ | $9.5_{\pm2.8}$ | $12.4_{\pm2.3}$ |
| | ETP | $15.3_{\pm7.1}$ | $5.2_{\pm3.4}$ | $42.1_{\pm13.1}$ | $14.1_{\pm8.6}$ | $9.0_{\pm2.5}$ | $3.3_{\pm1.0}$ |

**Robustness against perturbations.** We use the CIFAR10-C dataset in the same setup as in original work tHendrycks & Dietterich (2019). As FMNIST-C and SVHN-C are not available, we create them following the same procedure as in the original work. For FMNIST we adapt the procedure to gray-scale images by replicating the image three times for each channel. After applying the distortions, we map it back to the gray scale. Table 4 gives the numerical results used to create Figure 3 in the main paper.

