# OpenReview forum: "Evidential Turing Processes "
_ICLR.cc/2022/Conference — ICLR 2022 Poster_

### Official Review · Reviewer_cnXy · 2021-10-29

**Correctness:** 3
**Technical Novelty And Significance:** 3
**Empirical Novelty And Significance:** 3
**Recommendation:** 8
**Confidence:** 3

**Main Review:**

I don't think I understood the paper well enough to form a strong opinion about this model.
What lacks for me is a clear motivation for the following:
1. Why the Complete Bayesian Model is constructed this way. Is this the only plausible way to combine parametric and evidential approaches?
2. In the evidential DL papers that I've read, it seemed common to use entropies instead of variances to measure the spread of the distributions. Is there a reason for using variances here?
3. I wish there was some more intuition on what the interaction term in Eq.10 is doing. Model uncertainty and data uncertainty are quite easy to grasp: in the evidential DL literature they are usually visualized on a simplex (for Dirichlet priors), which makes them readily understandable. Is there a similar illustration that could explain the interaction term?
4. Also, how important is the interaction term?  I wish there were ablation studies that measure its effect in isolation.
5. I disagree with the approach of verifying the Main Hypothesis by showing that experimentally ETPs work better. These are non-trivial models that probably depend on a number of hyperparameters, so tuning ETP slightly better can have a greater effect on the performance compared to the effect of having a genuinely better model. A proper ablation is thus needed to account for this.
6. Definition 1, in my opinion, needs a better explanation. To me, it seems like any exchangeable process would have these properties. M', as I understand it, is a set of posterior parameters that can be computed from the prior parameters and the observations. What makes these properties so special that they define a Turing Process is unclear to me.
7. If we take an NP and encode the whole training set as a context, which we then use during test time, would that work? I mean, is it really necessary to have a rather complex memory mechanism of the Neural Turing Process?
8. I am underwhelmed by the experimental results: Table 1 is hard to understand, so it's not obvious why ETP is better. Also, if ECE is an important metric, which I assume is, then it would be best to explain it in the text.   Robustness experiments are pretty weird: I don't know how to interpet them or what was expected. If these experiments are included, then I think more than 4 lines of text are required to explain them.

Finally, I was surprised by the abrupt ending of the paper: I wish there was some sort of discussion. I can only assume that the paper was submitted in a rather unfinished state.


**Summary Of The Paper:**

The paper presents a complete Bayesian model that combines strengths of parametric and evidential Bayesian models.  An instance of this model is an Evidential Turing Process (ETP) -- a model in which the prior on the global variable that governs the likelihood function is input-dependent, and this dependence is established in a way similar to the Neural Turing Machines. Unlike Neural Processes (NPs), it doesn't need a context set during test time as the context is extracted from the training set and stored in a memory variable. Experiments show that  ETPs are somewhat better than BNNs and evidential NPs.

**Summary Of The Review:**

Given the points above, I think that the paper lacks clarity. As I mentioned earlier, I cannot form a strong opinion before I understand the motivation for the model and the experiments.

---

> ### Author Response · Authors · 2021-11-22
> **CBM construction justified, ablation table provided, interaction term re-explained, Table 1 simplified, other questions answered.**
>
> Thanks for helping us improve our work.
>
> 1. We revised Sections 2, 3, and 4 heavily and now provide a stepwise and theoretically-backed justification of our Complete Bayesian Model construction. It is indeed the optimal way to combine parametric and evidential approaches to inherit their desired uncertainty quantification properties. We encourage you to check out especially the beginning of the new Section 4 for details.
>
> 2. Following the original EDL work, we use the predictive entropy for out-of-domain detection. However, we base our analysis of prediction uncertainties on variance, as entropy does not provide an analogue of the law of total variance. Variance and entropy are closely linked anyway in discrete distributions.
>
> 3. We sympathise the confusion caused by the notion of "interaction term". In the new revision, we derive all uncertainty components from first principles and the corresponding term is now called "irreducible model uncertainty". Intuitively, it quantifies the lack of knowledge on the heteroscedastic noise.
>
> 4. The former "interaction term" and the new ""irreducible model uncertainty" term is relatively less important in achieving total calibration. However, we still find it useful to introduce the concept when characterising the uncertainty profiles of different Bayesian modeling approaches.
>
> 5. We now provide an ablation table in Table 1, where we detail how ETP can be broken down into other approaches by stepwise deactivating its components. There is no hyperparameter that is unique to ETP in our comparison list. Furthermore, we use the same hyperparameter tuning principles for all methods in our comparison.
>
> 6. The specialty of $M'$ is that it allows $p(\theta|Y')$ to incorporate context data via a user-defined external update rule. This makes it possible to detach the memory update from posterior inference, lifting the need for maintaining a context set to be able to do prediction at the test time.
>
> 7. An NP can fit the parameters of its context aggregator only if it uses the context observations to predict non-context training data "during training time". This means, it is technically not possible to train an NP with a context as large as the training set size. The case where the context is a subset of the training set is already handled in our experiment setup by the baseline ENP, which is a variant of NP that is equipped with additional properties to maximally challenge our proposed ETP. From the fact that ETP consistently improves over ENP, we conclude that our experiments empirically verify the necessity of the Turing Process component.
>
> 8. We simplified the presentation of Table 1 and Figure 3 in the new revision. We hope the superior performance of ETP over the baselines looks clearer now. We now introduce ECE at the top of Page 3 as part of the formal definition of "total calibration". We now clarify in Section 8 that we conduct the robustness experiments to evaluate how well models perform under gradual shift from the native domain. We perform the stress test by observing the in-domain calibration score ECE while the domain is shifting outwards.
>
> We added a summary paragraph to the conclusion section and improved the structure of the rest of it.

---

> > ### Comment · Reviewer_cnXy · 2021-11-25
> > **revision**
> >
> > Thank you for answering my questions and taking into account all the comments!
> > I think the paper has improved a lot, so I've increased my score.
> > I re-read everything up to Section 5 and skimmed the rest.
> > Some extra questions and suggestions:
> > - in the main hypothesis, it would be nice to explicitly say what the tasks 1,2,3 are. As I understand it now, you're referring to the tasks of quantifying the uncertainties due to model mismatch, class overlap and domain mismatch. Is that right?
> > - I think certain things could be made simpler or explained better, e.g. I don't quite understand the purpose of the Proposition on page 4.
> >  It is also quite unreadable and maybe should be placed in the appendix.
> > - Is this paper the first one to propose the "EDL as a latent variable model" view? I find it very interesting, and it seems to me the right way to do it. So have other papers presented this view already?

---

> > > ### Author Response · Authors · 2021-11-26
> > > **Thanks for your meticulous evaluation of our work! Here are our answers:**
> > >
> > > * That is right, what we meant in that sentence is indeed model misfit, class overlap, and domain mismatch. We will spell them out in the way you suggest.
> > >
> > > * We use the proposition to point to the fact that the Bayes risk of a classifier (Eq 4) and the second component of its predictive variance (Eq 9) are proportional: the minimum of two values summing to one, $1- \max f_{true}^{\pi}(x)$ and $\max f_{true}^{\pi}(x)$ from Eq 4, is proportional to their product, which is the variance of a categorical distribution taking up a specific value, i.e. $Var[y_*=k | x_*, \theta]$. This gives a mechanical explanation for why the second term on the r.h.s. of Eq 9 quantifies class overlap. Despite the commonplace allocation of the wording "data uncertainty" for this term, we are not aware of prior work that derives it formally from learning-theoretic concepts. We are happy to move the proposition to the appendix and replace it with a verbal explanation in the main paper which is derived from our answer here.
> > >
> > > * We are not aware of any published work that proposes the LVM view for EDL. With some further search based on your inquiry, we found the unpublished ArXiv post below, which proposes the same formulation without framing it explicitly as variational inference of a LVM: https://arxiv.org/pdf/1811.07308.pdf. We will cite this work in Page 4 at the end of the sentence: "Although presented in the original work as...".
> > >
> > > In case you have further questions, we are very happy to answer them.

---

### Official Review · Reviewer_zTRM · 2021-11-01

**Correctness:** 2
**Technical Novelty And Significance:** 2
**Empirical Novelty And Significance:** 3
**Recommendation:** 5
**Confidence:** 3

**Main Review:**

This paper has several strengths to acknowledge:
- I really appreciate the top-down approach of starting with general principles, which are progressively transformed into an actual implementation which obtains competitive performance.
- The category of complete Bayesian models deserves its name of "complete" as it covers a very large range of models (see Section 4).
- Creating a unified approach to uncertainty quantification is a valuable and ambitious goal.

But I am unfortunately not yet convinced by the validation of the main claims of the paper:

*"**Main Hypothesis.** A Bayesian model which is capable of quantifying both uncertainties in isolation, as well as their interaction explicitly is a guiding principle for predictors that can succeed in in-domain calibration and out-of-domain detection simultaneously."*

*"**Our Solution Proposal.** We conjecture that a prior can effectively depend on an input if it receives context information during training time, stores it into an external memory, and retrieves relevant parts of it using the conditioned input example as a query."*

- The experimental validation is undermined by the very small models, self-described as "LeNet5-size", with only a few layers. The reported prediction error seem unreasonably high (15+% on CIFAR10 for instance). Please clarify any misunderstanding concerning this point.
- There is no experimental validation for the choice of a Turing process. What about the alternative of using a simple neural network predicting $\pi$ from input $x$ to amortize the prior $p(\pi|\theta, x)$?
- Despite a lot of math in the paper, the theoretical justification that is actually relevant seems rather limited. A large part of the equations appear to be somewhat decorative. For instance:
  - As far as I understand, the expressions of variational free energy (4) (5) (9) (13) (17) (19) don’t support any argument of the paper
  - As a second example, Definition 1, (18) and (26) are not contributions of this paper, but describe already existing components (Turing processes, neural processes, the attention mechanism)



**Summary Of The Paper:**

This paper introduces the category of complete Bayesian models, with the goal of providing both in-domain uncertainty quantification (that is, calibration) and and out-of-domain detection. Concretely, the authors develop the evidential Turing process as an implementation of complete Bayesian model. The approach is experimentally validated by experiments on FMNIST, CIFAR10 and SVHN, showing that the evidential Turing process is competitive on all criteria.

**Summary Of The Review:**

I really appreciate the top-down approach of the paper. Unfortunately, I am neither really convinced by the experimental validation which considers too small models, nor by the theoretical presentation that sometimes feel a bit decorative. Hence my weak reject recommendation, with confidence rated at 3 because I am not familiar with evidential Bayesian models. I am looking forward to discussion with the authors to clarify any misunderstanding.

---

> ### Author Response · Authors · 2021-11-22
> **New results provided, theoretical justification included, some redundant content removed**
>
>
> It is correct that CIFAR10 error is around 15% as a result of the LeNet5 architecture. We adopted this architecture to the setup used in the original EDL work (Sensoy et al., 2018). We now report results on CIFAR100 using the ResNet18 architecture: 10-fold increase in class count and an 18-layer architecture. We also report results on the IMDB data set with a recurrent neural net design. ETP still outperforms its counterparts.
>
> We agree with the redundancy of some information we provided and the necessity for more theoretical rigor in some parts of our material. We removed the repetitive introductions of variational free energies and used the space for deepening the theoretical justification of the CBM and ETP design choices, as well as their match with a more formal definition of total calibration in Sections 2, 3, and 4.
>
> Definition 1 is indeed a contribution of our work. We are the first to use the NTM design in conjunction with neural processes and to formulate the resultant model as a new kind of stochastic process. The closest match in prior work to ours is the Attentive Neural Process (Kim et al., 2019), which builds an attention network into a neural process but without the Turing machine design.
>
> The simple neural net predicting $\pi$ from input with inactive memory boils down to vanilla EDL, which is already in our comparison list. Activating $\theta$ in this setup would make it only a training-time regularizer, which is then discarded at the test time where only $q(\pi | x)$ is used. This would be an inference approach where the approximate posterior and the true posterior do not have the same set of random variables.

---

> > ### Comment · Reviewer_zTRM · 2021-11-29
> > **Response to Authors**
> >
> > Thanks for clarifying my misunderstandings about Definition 1 and about the vanilla EDL baseline, and for removing the variational free energies formulas.
> > - Concerning model size, making a larger scale experiment on CIFAR100 is a good start, but this does not address the fact that 15% prediction error on CIFAR10 (and 30% on CIFAR100) is not competitive. It is hard for me to comment on the authors's point that original EDL work (Sensoy et al., 2018) use the same model sizes, as I'm not very familiar with the EDL literature (as mentioned in my original review).
> > - The rebuttal version of the submission is significantly different from the version I originally reviewed, in particular Section 2, 3 and 4. To me, these modifications are beyond the scope of what can reasonably be expected in a rebuttal phase, and I would find it desirable to reach more stability in the writing. I would also encourage the authors to stick with the original top-down approach of starting with general principles, which are progressively transformed into an actual implementation which obtains competitive performance. As mentioned in my original review, I think this was a strength.
> >
> > Considering these aspects, I will stick to my weak reject recommendation, with mid-range confidence because I am not very familiar with the EDL literature.

---

### Official Review · Reviewer_L3yb · 2021-11-02

**Correctness:** 4
**Technical Novelty And Significance:** 3
**Empirical Novelty And Significance:** 2
**Recommendation:** 6
**Confidence:** 2

**Main Review:**

Even though I am far from an expert in the relevant literature, I find this paper very well written and was able to follow most of the presentation. It clearly introduces the relevant concepts and explains the underlying motivation for the proposed methods.

I find the experimental section in the main text a bit brief. I personally find this justifiable as allows the manuscript to explain and motivate the proposed method in sufficient detail. I do want to encourage the authors to be a bit more restraint with marking the best method by bold letters in Table 1. If two or methods lead to the same results within one (or even three) standard deviations, one should either mark all of these methods as performing best or none of them.

A few minor suggestions:
-Equation (1) ends with both a full stop and comma.
-Equation (4) misses a "(" in the subscript of the expectation value




**Summary Of The Paper:**

The manuscript proposes Evidential Turing Process which is a combination of Neural Turing Machines and Neural Processes. The authors theoretically motivate and empirically demonstrate that their proposed method leads to better in-distribution uncertainty estimation as well as improved out-of-distribution detection.

**Summary Of The Review:**

I find the paper very well written. The proposed method appears to be theoretically well-motivated and, empirically, leads to improved robustness.

I want to stress however that I have only a very limited overview of the relevant field of research and I can thus not confidently state a definite opinion, especially concerning the novelty of the manuscript.

---

> ### Author Response · Authors · 2021-11-22
> **Best performers highlighted in bold, further experiment details provided, typos removed**
>
> Thanks for helping us improve our work.
>
> Following your well-justified suggestion, the new Table 2 highlights all best performing models in bold within a three standard deviations interval.
>
> We modified Section 8 towards more to-the-point descriptions of experiment details. We introduced Table 1 that demonstrates an ablation plan for our experiment setup. We also explained our performance evaluation metrics in more detail at the top of page 3 when introducing a formal definition of "total calibration". We hope these additional pieces of information have made our experiments more accessible to the readership.
>
> Thanks for pointing out the typos. Equations 1 and 4 are replaced by other content in the new revision. Thus, the typos should no longer exist.

---

### Official Review · Reviewer_UsNy · 2021-11-05

**Correctness:** 4
**Technical Novelty And Significance:** 3
**Empirical Novelty And Significance:** 3
**Recommendation:** 6
**Confidence:** 2

**Main Review:**

Strengths:

 - The idea is interesting and well motivated.
 - The experimental results highlight the improvement over the baselines in terms of a series of metrics.
 - The paper is generally well-written.

Comments:

 - I believe the paper would benefit from experiments on a more diverse set of  (not necessarily very large) architectures (e.g., ResNet20).
 - It would be interesting to see results on real-world data beyond images, e.g.,  IMDB sentiment classification.
 - I appreciate the illustrative examples in 1-D and 2-D. It would be interesting to see a similar experiment on a small-scale image dataset (e.g., MNIST).

**Summary Of The Paper:**

The paper introduces a “Complete Bayesian Model” that combines the benefits of Parametric Bayesian Models and Evidential Bayesian Models. To this end, the authors propose a technique to learn an input-dependent prior. The experiments showcase the efficacy of the method compared to the baselines.

**Summary Of The Review:**

Overall, this is an interesting work. I lean towards acceptance, especially if the comments that were raised are addressed.

---

> ### Author Response · Authors · 2021-11-22
> **Suggested experiments are done. Results are in favor of our method.**
>
> Thanks for helping us improve our work. Having agreed with your suggestion, we now provide results on the CIFAR100 data set with ResNet18 and IMDB sentiment recognition data set with an LSTM architecture. As visible in the first and last columns of Table 2 in the new revision, our ETP is the only model that ranks among the best performance according to all success criteria simultaneously.
>
> Unfortunately we ran out of space and time to provide an additional illustration on the MNIST data set. The two illustrations provided in the original submission are now in appendix.

---

### Author Response · Authors · 2021-11-22
**New revision addresses all issues raised by the reviewers**

We thank all reviewers for their careful read on our work and the high-quality feedback they gave about it. Their comments helped us realize some of its weaknesses, discover some of its initially overlooked strengths, and above all deepen our own understanding about it. We have uploaded a heavily revised new version of our work, which we believe to address all issues raised by the reviewers.

To summarize:

i) We realised that some of the key messages of our work were blurred due to the absence of more formal problem statement than the one given in the original submission. In a completely rewritten Section 2, we now provide a logical development starting from the Bayesian decision theory for multi-class classification and going stepwise towards a formal definition of total calibration, the ultimate goal we aim to achieve.

ii) We also rewrote Section 3 by deriving the key uncertainty characteristics of parametric and evidential approaches from the formal concepts developed in the new Section 2.

iii) We start Section 4 with a mapping from the uncertainty characteristics discussed in Section 3 to the properties of total calibration, which then paves the way to our Complete Bayesian Models proposal.

iv) At the top of page 7, we now provide an ablation table that shows how other state-of-the-art approaches recover when we deactivate the components of our ETP one at a time. We hope this table clarifies the fact that the results we report in the original submission already reflect a systematic ablation study.

v) Upon the rightly pointed out request of multiple reviewers, we now report results on two new data sets and architectures: IMDB sentiment recognition data set using an LSTM architecture and CIFAR100 using a ResNet100 architecture. The results of both experiments are in favor of the main claims of our work.

vi) We added a summary paragraph to the conclusion and made minor modifications in the rest of it, agreeing with another reviewer request.

We sincerely believe that the revised version of our work has significantly improved on the original submission thanks to great input we received from the reviewers. We are grateful for their effort.

---

### Decision · Program_Chairs · 2022-01-20

**Decision:**

Accept (Poster)

**Comment:**

It seems the reviewers are in an agreement that the work seems interesting, well motivated, and results are meaningful. The main complaints or issues that remain is the amount of rewriting involved, which might be hard for the reviewers to track, and maybe question regarding the results given for e.g. the choice of architectures for CIFAR-10 making numbers harder to interpret.

However, I do see that the quality of the manuscript is quite good (and multiple reviewers commented on this), and the idea seems natural to me. I think the results, especially with the addition of CIFAR-100 and IMDB seem sufficient, and given the overall positive feeling of the reviewers over the work with no major concerns, I am happy to recommend this paper for acceptance.